# Research on the Development and Application of a Deep Learning Model for Effective Management and Response to Harmful Algal Blooms

**Jungwook Kim, Hongtae Kim, Kyunghyun Kim**  **and Jung Min Ahn ***

Water Quality Assessment Research Division, Water Environment Research Department, National Institute of Environmental Research, Incheon 22689, Republic of Korea; kjw1128@korea.kr (J.K.); htkim8@korea.kr (H.K.)
* Correspondence: jahn@korea.kr; Tel.: +82-32-560-7490

**Abstract:** Harmful algal blooms (HABs) caused by harmful cyanobacteria adversely impact the water quality in aquatic ecosystems and burden socioecological systems that are based on water utilization. Currently, Korea uses the Environmental Fluid Dynamics Code-National Institute of Environmental Research (EFDC-NIER) model to predict algae conditions and respond to algal blooms through the HAB alert system. This study aimed to establish an additional deep learning model to effectively respond to algal blooms. The prediction model is based on a deep neural network (DNN), which is a type of artificial neural network widely used for HAB prediction. By applying the synthetic minority over-sampling technique (SMOTE) to resolve the imbalance in the data, the DNN model showed improved performance during validation for predicting the number of cyanobacteria cells. The R-squared increased from 0.7 to 0.78, MAE decreased from 0.7 to 0.6, and RMSE decreased from 0.9 to 0.7, indicating an enhancement in the model's performance. Furthermore, regarding the HAB alert levels, the R-squared increased from 0.18 to 0.79, MAE decreased from 0.2 to 0.1, and RMSE decreased from 0.3 to 0.2, indicating improved performance as well. According to the results, the constructed data-based model reasonably predicted algae conditions in the summer when algal bloom-induced damage occurs and accurately predicted the HAB alert levels for immediate decision-making. The main objective of this study was to develop a new technology for predicting and managing HABs in river environments, aiming for a sustainable future for the aquatic ecosystem.

**Keywords:** harmful algal blooms; deep neural network; synthetic minority over-sampling technique; number of cyanobacteria cells; HAB alert levels



## 1. Introduction

Due to river dredging and the construction of multi-function weirs as part of the four major river restoration projects, riverine environments in South Korea (hereinafter Korea) underwent drastic changes within a short period of time [1]. However, they are deteriorating owing to the changes in meteorological and hydraulic conditions due to recent climate change. Among the changes in the riverine environments, increased hydraulic residence time is a major factor exacerbating harmful algal blooms (HABs) in summer every year. Furthermore, rainfall causes changes in the hydrodynamic conditions of the river, leading to sediment resuspension. The resuspension of sediment can re-suspend the bed sediments that are home to large nutrient loads in the river [2]. Excessive nutrient pollution causing eutrophication triggers the proliferation of HABs and poses a negative impact on the environment. In the world's largest inland lake, the Caspian Sea, there are also challenges due to the increase in Chlorophyll-a (Chl-a) during warm months, and the rate of eutrophication is increasing [3]. The eutrophication in Lake Erie continues to worsen, and if this trend continues, it is projected that methane emissions from the lake will account for 38 to 53% of the greenhouse gas emissions from fossil fuels [4]. From 2012 to 2023, based on the analysis of the total phosphorus (TP) observed

by the Ministry of Environment in Korea, it was found that TP in the Nakdong River exceeds the Organization for Economic Cooperation and Development (OECD) standard of 0.035, indicating a eutrophic condition. Eutrophication in the Great Lakes in North America, Lake Taihu in China, and Lake Victoria in Africa is indeed accelerating. Additionally, the number of water bodies, including lakes and rivers, experiencing eutrophication is increasing [5]. With the exacerbation of eutrophication and climate change, the occurrence of toxic algae is increasing globally [6].

In Korea, algae start growing during spring when the water temperature begins to rise and stops in late autumn. Cyanobacterial blooms are the dominant species in summer. In terms of the net zero of the Sustainable Development Goals (SDGs), algae have the potential to be utilized as an energy resource that minimizes greenhouse gas emissions [7]. However, HABs produce toxic substances such as microcystins, negatively impacting the aquatic ecosystem. Therefore, it is necessary to proactively manage and prepare for the occurrence of HABs through prediction rather than utilizing them as bioenergy sources.

They rapidly proliferate in large quantities and develop into HABs when solar radiation increases and water temperature is high [8]. *Microcystis* is a major cyanobacterium that can cause harmful algal blooms not only in foreign countries but also in Korea, including the Nakdong River. In particular, most HABs are caused by *Microcystis* in Korea. Several studies have reported that an increase in toxic cyanobacteria, such as *Microcystis*, can lead to elevated toxins in rivers, causing contamination of drinking water sources and adverse effects on aquatic organisms. A research team from Tibet University and North Carolina State University revealed that climate change is causing shifts in the composition of cyanobacterial communities in lakes. Particularly, it was observed that *Microcystis*, as a dominant species, has increased in response to climate change, leading to an elevation of toxins in lakes [9]. *Microcystis* is known to produce toxins called microcystin [10,11]. In other words, if the abundance of *Microcystis*, a toxic cyanobacteria, increases, the toxins in the river also increase. According to [12], the Nakdong River in Korea experiences HABs caused by *Microcystis* during the summer season due to the construction of eight weirs that have led to water stagnation and thermal stratification. Furthermore, it is reported that during the summer season, the majority of proliferating cyanobacteria are *Microcystis*, and as their abundance increases, toxins also rise. Problems caused by HABs have a direct impact on the lives of people who drink treated river water.

Korea uses EFDC-NIER, a physics-based model, to predict the number of cyanobacteria cells. Next, an alert is issued according to the HAB alert criteria, which is divided into levels from 0 to 4 based on the number of cyanobacteria cells. Physics-based models are advantageous because the actual river environment can be built into the model; therefore, they are widely used for algae prediction in various countries [13–15]. The prediction accuracy of physics-based models varies with parameters [16–18]. However, building input data and calibrating the parameters require extensive time and knowledge of the model [19].

An artificial intelligence (AI)-driven deep learning model can solve the problem of a physics-based model. Data-based models may be used for solving the problem of excessive time requirements of a physics-based model for prediction and parameter correction, provided that sufficient high-quality data are available as input data. This is because it learns various types of prediction information and makes predictions with the trained algorithm. AI-driven deep learning models have developed rapidly over the past 20 years owing to advances in computers and other hardware and demonstrate excellent prediction performance in diverse fields such as climate, atmosphere, economy, water resources, and water quality [20]. Hardware advances have led to the development of AI-driven deep learning models, such as artificial neural networks (ANNs), which can handle a large amount of computation, including complex problems and big data that are difficult to solve mathematically [21]. Different types of ANNs have been developed depending on their purpose. These include deep neural networks (DNN) for processing complex data, convolution neural networks (CNN) for extracting image features, recurrent neural networks (RNN) for learning time-series or sequential data, long short-term memory (LSTM), and gated recurrent

unit (GRU). Thus, the users can select a model according to various conditions, such as the characteristics of the data and the analysis objectives. Unlike water quality, algae are living organisms and have diverse life patterns even under identical environmental conditions, making data-based HAB prediction a very challenging field of research. Therefore, complex learning is required to predict algae conditions, and research on harmful algae prediction based on ANNs is underway. ANNs have been widely used for predicting algae in the past, and their application is increasing as ANNs-based algorithms have advanced.

Jeong et al. [22] predicted *Microcystis aeruginosa* bloom dynamics using evolutionary computation and ANN in the Nakdong River, Korea. Velo-Suarez and Gutierrez-Estrada [23] predicted *Dinophysis acuminata* blooms for weekly intervals using ANN. Maier et al. [24] conducted a study where they constructed an ANN model to predict *Microcystis* biomass in rivers and evaluated its predictive performance. The neural network-based model provided valuable insights into the dynamics and potential risks associated with algal blooms in river environments. Indeed, the factors influencing algal blooms are diverse and complex, encompassing various water quality and meteorological variables. For highly complex datasets, many researchers are utilizing upgraded models based on DNN to predict water quality [25]. The research on algae prediction is actively conducted using various models that have been developed for different purposes based on the DNN algorithm. Pyo et al. [26] utilized a CNN model to predict the spatial and temporal distribution of harmful cyanobacteria, specifically *Microcystis*. Ni et al. [27] employed an LSTM model to estimate Chl-a concentration and used it to predict the occurrence of CyanoHABs. Indeed, there are also cases where a new DNN model has been built by combining DNN with statistical techniques to predict algal blooms. A new model combining DNN, data decomposition, and fuzzy clustering was proposed to predict water quality factors influencing algal blooms [28]. An SAE-DNN model, combining the stacked autoencoder (SAE) technique with DNN, was developed to estimate the concentration of phycocyanin in cyanobacteria [29]. In some regions, there is a shortage of data or data imbalance, and recent research is being conducted to address this issue. To protect water resources from contamination, Yang et al. [30] constructed a model called CNN-LSTM with attention to predicting water quality variables such as pH and ammonia nitrogen. In this study, missing data were interpolated using the linear interpolation method. Despite the advancements in AI-based deep learning techniques, there has been recent research focused on improving the predictive power of models with limited training data. For example, novel frameworks based on the multivariate distributions (MVD)-based virtual sample generation (VSG) method have been developed to generate virtual data samples and train the DNN model using these synthesized data. The results of these studies are compared with real data to assess their applicability [31]. In this research, it is mentioned that future studies are crucial and important in creating optimal virtual datasets.

AI-driven deep learning models are rapidly advancing and widely used in predicting natural environmental data. Indeed, there is a debate regarding whether this model can replace traditional physics-based models. According to Sit et al. [32], they argue that the automation of hydrological modeling using AI-based deep learning technology can raise ethical concerns in disaster management and public planning. AI-driven deep learning models are indeed simple and have good predictive power. However, it is difficult to determine the learning structure of these models and understand or evaluate the uncertainty of simulations [33]. Therefore, there is an opinion that if the suitability of simulations cannot be predicted and reviewed, the responsibility for the results may also be unclear [34,35]. For example, if the prediction results are inaccurate due to weather anomalies such as heavy rainfall, it can lead to incorrect flood predictions. This can result in potential risks to human lives and property damage [36]. Despite these limitations, the demand for AI-driven deep learning technology is continuously increasing in various fields worldwide [37,38]. Therefore, research on this technology is essential to keep up with this trend.

This study aims to develop an AI-driven deep learning model that can predict algae conditions to effectively respond to algal blooms. As mentioned above, since algae are living organisms, predicting their future behavior requires more knowledge and effort than

in other fields. In particular, rivers are used as a source of drinking water in Korea, and accurately predicting algae conditions is important for the health of citizens and safe water use. From the perspective of researchers predicting algae conditions, it is crucial to apply various HAB prediction models to respond to algal blooms when determining the alert level. To this end, we constructed an AI-driven deep learning model that has proven its prediction accuracy in various fields and evaluated its applicability.

The research objectives of this study are as follows:

1.  The relationship between harmful algae and water quality is nonlinear. Of the various ANN-based algorithms, DNN, in particular, has the advantage of enabling nonlinear combinations between input variables. Additionally, it is a powerful tool for modeling complex systems [39,40]. Furthermore, to address the data imbalance in the data of cyanobacteria cell counts in the study area, a model combining DNN with SMOTE was proposed, and its applicability was evaluated. In this study, the DNN algorithm was used to simultaneously predict the number of cyanobacteria cells and the HAB alert levels. To allow for the prediction of both continuous and categorical data, the DNN algorithm was improved to support multiple outputs in this study.
2.  In terms of prediction accuracy, we evaluated the results of the DNN algorithm, which predicts cyanobacteria cells and HAB alert level, to assess their applicability in the field and recommend research directions for future AI-based HAB prediction research.

## 2. Materials and Methods

### 2.1. Study Area

In this study, data-based HAB prediction was performed at Changnyeong-Haman weir, located downstream of the Nakdong River (Figure 1). This site is affected by cyanobacterial blooms formed by *Microcystis* every summer. In addition, this region is severely affected by HAB-induced damage. In particular, downstream of the Nakdong River, a part of the study area is used as a source of drinking water by residents of nearby cities such as Busan, the second-largest city in Korea. In this area, systematic algal bloom management and pre-emptive response are vital for the public's health and safe water use.

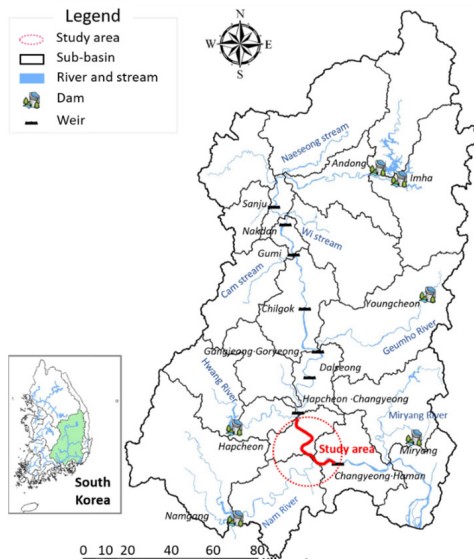

**Figure 1.** Study area.

### 2.2. Water Quality, Algae, and HAB Alert Levels Data

After the construction of multi-function weirs in 2012, the government measured the water quality and HABs in weir sections at weekly intervals to respond to and manage HABs. The occurrence of HABs is affected by water quality and organic matter. The water quality and organic matter items measured in the study area of Changnyeong-Haman

weir were water temperature, pH, dissolved oxygen (DO), the 5-day biochemical oxygen demand ($BOD_5$), chemical oxygen demand (COD), suspended solids (SS), total nitrogen (TN), total phosphorus (TP), and total organic carbon (TOC), and the algae items were chlorophyll-a (Chl-a) and the number of cyanobacteria (Table 1). Cyanobacterial blooms proliferate in high-insolation and high-temperature conditions and are measured only in summer. Due to the nature of cyanobacterial blooms to proliferate only during a specific period in summer, it is difficult to obtain sufficient input data necessary for data-based prediction models to analyze and learn the correlations between cyanobacterial blooms and water quality (temperature and organic matter) associated with their proliferation and growth. In general, the predictive power of data-based models varies from one prediction model to another depending on the quantity and quality of the training set. The data were collected from 2012 to 2022, excluding the item for the number of cyanobacteria cells. The data from 2012 to 2021 were used for training and validation, and the data from 2022 were used to evaluate the prediction accuracy of the constructed model. Cyanobacteria mainly occur and proliferate during the summer season when the water temperature is high, and there is a very strong correlation between these two factors (Figure 2).

**Table 1.** Range, averages, and standard deviations of variables related to water quality and algae used for training and validation (2012–2021).

| Variables | Changnyeong-Haman Weir | |
|---|---|---|
| | MIN–MAX [1] | AVG ± SD [2] |
| Water temperature (°C) | 2.0–35.1 | 16.6 ± 8.6 |
| pH | 6.7–9.7 | 8.1 ± 0.6 |
| DO (mg/L) | 6.4–21.5 | 11.2 ± 2.5 |
| $BOD_5$ (mg/L) | 0.7–5.6 | 2.3 ± 0.8 |
| COD (mg/L) | 3.9–12.8 | 6.3 ± 1.1 |
| SS (mg/L) | 1.6–72.0 | 10.1 ± 6.9 |
| TN (mg/L) | 1.157–5.483 | 2.767 ± 0.771 |
| TP (mg/L) | 0.013–0.174 | 0.050 ± 0.029 |
| TOC (mg/L) | 2.6–11.1 | 4.3 ± 0.9 |
| Chl-a (mg/m$^3$) | 2.2–134.4 | 26.6 ± 19.3 |
| Number of Cyanobacteria (cells/mL) | 13,557–715,993 | 13,557 ± 51,435 |

Notes: [1] Minimum to maximum. [2] Average and standard deviation.

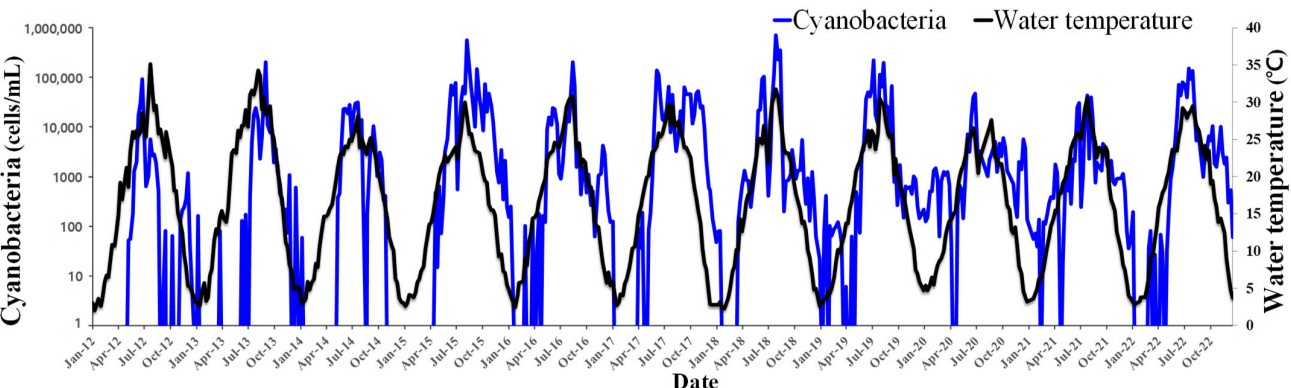

**Figure 2.** Observed data on water temperature and cyanobacteria in the study area.

In Korea, HAB alerts are issued from levels 0 to 4 according to the number of cyanobacteria cells. In the dataset collected from 2012 to 2022, there were 314 cases for level 0, 126 for level 1, 93 for level 2, 17 for level 3, and 0 for level 4 (Table 2). Therefore, excluding level 4, which had no data, the dependent variable data were constructed with four alert categories (levels 0 to 3).

**Table 2.** HAB alert system in Korea.

| Level | 0 | 1 | 2 | 3 | 4 |
|---|---|---|---|---|---|
| Cyanobacteria (cells/mL) | <1000 | $\geq$1000 <10,000 | $\geq$10,000 <100,000 | $\geq$100,000 <1,000,000 | $\geq$1,000,000 |
| Counts | 314 | 126 | 93 | 17 | 0 |

Selecting appropriate input variables is important to ensure prediction accuracy. In particular, water temperature, pH, DO, BOD, COD, SS, TN, TP, TOC, and Chl-a are important for algal growth, among which water temperature, DO, and TP are reported to have the highest impact [41].

*2.3. DNN Algorithm*

Deep neural networks (DNN) are a type of ANN, an aspect of AI that focuses on mimicking the human learning approach to acquire specific knowledge. Most real problems have nonlinear characteristics. The amount of computation could be reduced, and the nonlinear problems can be solved by adding a backpropagation algorithm and two or more hidden layers. Learning various data, particularly big data, requires many hidden layers. However, an increase in the number of hidden layers is associated with the vanishing gradient problem, in which the errors in the output layer are not transmitted to the input layer, and the exploding gradient problem, in which the values are updated rapidly, thus making it difficult to train the neural network. Both vanishing and exploding gradient problems were solved by applying an unsupervised pre-training method, which reduced the errors at the output layer by preventing over-learning caused by the increase in hidden layers while learning from big data [42].

To construct an optimal DNN model, the number of hidden layers and node distribution for each layer must be set. Because the variables used for training and validation are weekly data, the data is relatively insufficient. To prevent a decrease in the learning rate due to the lower availability of water quality data, the hierarchical structure of the DNN model was appropriately designed.

*2.4. Method for Analyzing the Predictive Performance*

For training and validation sets, the water quality data from 2012 to 2021 were used, whereby the training and validation periods were set at the ratio of 8:2. We calculated the values of $R^2$ (Equations (1)–(3)), mean of all absolute errors (MAE) (Equation (4)), and root mean square error (RMSE) (Equation (5)) and estimated the accuracy of the models. $R^2$ indicates how well the predictive value explains the measured value, and the higher the value of $R^2$, the higher the prediction accuracy [43]. MAE is the difference between estimated and measured values [44], and RMSE is an error metric used to assess the difference between the estimated and measured values [45]. The lower the MAE and RMSE values of a prediction model, the higher the prediction performance of the model.

$$R^2 = 1 - \left( \frac{SSR}{SST} \right), \tag{1}$$

$$SST = \sum_{i=1}^{n} (y_i - \overline{y}_i)^2, \tag{2}$$

$$SSR = \sum_{i=1}^{n} (y_i - \hat{y}_i)^2, \tag{3}$$

$$MAE = \sum_{i=1}^{n} |y_i - \overline{y}_i|, \tag{4}$$

$$\text{RMSE} = \sqrt{\left(\frac{1}{n}\right)\sum_{i=1}^{n}(y_i - \hat{y}_i)^2}. \tag{5}$$

## 3. Construction of Deep Learning Model and Prediction Results

### 3.1. Data Preprocessing

Generally, when training a data-based prediction model, the range of data values must be standardized. For example, TP ranges from 0 to 1, whereas the number of cyanobacteria cells ranges from 0 to hundreds of thousands (Table 1). If the scale difference between the values of each feature is too large in such a case, then a strong one-sided bias may occur, which can degrade the learning ability. Therefore, the data must be preprocessed before training. Prior to training, we standardized the magnitude of individual data by adjusting the scale difference between variables through normalization. Even when normalizing data, the distribution of each variable should not be modified; therefore, we selected the normalization method MinMaxScaler, which converts the values of each data variable with different maximum values to values between 0 and 1 to adjust their scale. Equation (6) shows the conversion equation.

$$\text{MinMaxScaler}(x) = \frac{x - x_{min}}{x_{max} - x_{min}}. \tag{6}$$

In contrast to the conventional analysis of algal time series data such as cyanobacteria, this study constructed a DNN model with a multi-output structure to simultaneously predict HAB alert levels. When predicting categorical data such as HAB alert levels, the proportion of each class is important. When predicting imbalanced categorical data where certain classes have skewed data, the prediction results can be distorted [46,47]. This is because the prediction model may be trained based on the majority of categories in the dataset. As shown in Table 2, it can be observed that the data are skewed towards level 0. Therefore, if these data are trained, the prediction results may be distorted and biased towards level 0.

There are techniques for handling imbalanced data, such as undersampling and oversampling. Undersampling is a technique to address imbalanced data by keeping only significant data. Applying undersampling to HAB alert levels data can result in a significant loss of the entire dataset and the loss of important normal data. This can lead to a shortage of usable HAB alert levels data for training, which may result in poor training. Oversampling is a technique used to handle imbalanced data by replicating data from the minority class based on a predetermined ratio. By increasing the dataset in this way, the proportion of minority class data is also increased, helping the training model handle imbalanced data. Deep learning analysis generally requires a large amount of data, so oversampling is commonly used to handle imbalanced data effectively.

In this study, the synthetic minority oversampling technique (SMOTE) algorithm was applied. SMOTE is a popular algorithm used to address the class imbalance in deep learning. In class imbalance, the number of samples in one class is much smaller than the number of samples in another class, which can lead to poor model performance. SMOTE is an oversampling technique that generates synthetic data by creating additional instances of minority class samples, thereby balancing the dataset. This is carried out by randomly selecting a minority class sample and finding its k-nearest neighbors. Then, new samples are synthesized by interpolating between the minority sample and its neighbors [48]. The amount of interpolation is controlled by a parameter that specifies the degree of oversampling required.

### 3.2. Hidden Layers and Nodes for Optimal DNN Model

The number of input and output nodes is determined by the type of input and output, but there is no set formula for determining the number of hidden layers and nodes [49]. However, there are some criteria for determining the optimal number of hidden layers

and nodes. In general, the complexity of the problem, the amount of data, and hardware and time are commonly considered major factors in determining the optimal number of hidden layers and nodes. In terms of the complexity of the problem, it depends on the functionality or application, so the structure of the DNN network should be referenced based on similar studies. The amount of data is an important criterion for determining the optimal number of hidden layers and nodes [50]. When the amount of data, such as weekly data, is not sufficient, having too many hidden layers can cause problems such as poor training or overfitting. In addition, if the number of hidden layers and nodes is too large, the execution time may become too long [51,52]. This study determined the optimal number of hidden layers and nodes through trial and error.

When comparing various combinations of hidden layers and nodes, the training accuracy was relatively higher in cases where the number of hidden layers was small, and the number of nodes was large. Increasing the number of hidden layers resulted in a decrease in training ability, which is believed to be due to the relatively small amount of water quality and algae data used for analysis. On the other hand, since various input data were analyzed simultaneously, models with a relatively large number of nodes had better training ability. Furthermore, considering the model's execution time and hardware, a DNN with two hidden layers and 50 nodes was determined to be the optimal structure for this study (Figure 3). In DNN learning, overfitting may occur by overlearning only certain explanatory variable features. To prevent this, an appropriate dropout should be set [53]. Dropout refers to the ratio of nodes uniformly dropped from the units of each hidden layer, and a dropout ratio of 0.2 is considered the optimal ratio [54].

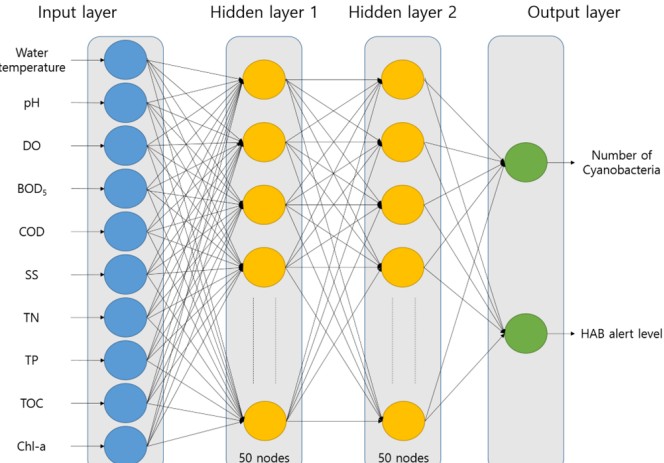

**Figure 3.** Diagram of DNN architecture.

Next, a Rectified Linear Unit (ReLU) activation function was used to transform input data into output signal. The other activation functions, sigmoid and tanh, have problems such as a decrease in learning rate and vanishing gradient. The ReLU activation function is a widely used function that solves the problems of sigmoid and tanh, yields high performance, and has a very simple structure [49]. However, when most of the input values are negative, it is difficult to conduct backpropagation by the gradient. As a result, the input values are not updated, and their use becomes limited. As the water quality data used as input values in this study does not have negative values, the ReLU function can be used.

Epoch refers to the number of times the entire dataset passes through the neural network. Setting an appropriate number of epochs is important for the model's predictive power. The method for determining the optimal number of epochs involves selecting the number at which the performance improves and then decreases due to overfitting in the validation set [55]. Here, the performance is confirmed through the loss function. If the number of epochs is too small, the model's predictive power decreases, and if it is too large, the model may become overfitted. In the training set, the loss decreased rapidly until

2000 epochs, after which the loss values were similar. In the validation set, the minimum loss occurred around 5000 epochs, after which the loss gradually increased. This indicates that when using values larger than 5000, the prediction performance deteriorates due to overfitting. Therefore, setting the number of epochs to 5000 is reasonable. To minimize the loss function during DNN training, the Adam optimizer was used. It is an algorithm that combines momentum and root mean square propagation (RMSProp) and has an adaptive learning rate depending on the amount of change in the curvature of the former landscape. It is the most widely used optimizer for deep learning models because of its good performance in different neural networks with a very wide range of architectures [56]. When applying Adam, the mean absolute error (MAE) was used as the loss function. In particular, if the optimizer is selected as Adam, the loss value can be minimized during DNN training.

### 3.3. Improvement of DNN Architecture for Multi-Output of Continuous and Categorical Data

The DNN structure for multi-output problems is similar to that of a general DNN, with only the number of nodes and activation functions in the output layer potentially differing. In multi-output problems, the number of nodes in the output layer is set to match the number of output variables, and the activation function may vary depending on the characteristics of each output variable. For example, in regression problems where continuous variables are output, a linear activation function is used in the output layer. On the other hand, in classification problems where categorical variables are output, a softmax activation function is used in the output layer. Therefore, the DNN structure for multi-output problems is designed by adjusting the activation function and the number of nodes in the output layer according to the characteristics of the output variables.

In this study, a multi-output structure targeting the continuous data of cyanobacteria cells and the categorical data of HAB alert levels needed to be constructed. The activation function should be selected as "linear", and the output node should be set to 1 when constructing the output layer for continuous data such as cyanobacteria cells. Additionally, the activation function should be selected as "softmax", and the output node should be set according to the number of classes when constructing the output layer for multi-class categorical data such as HAB alert levels (Figure 4).

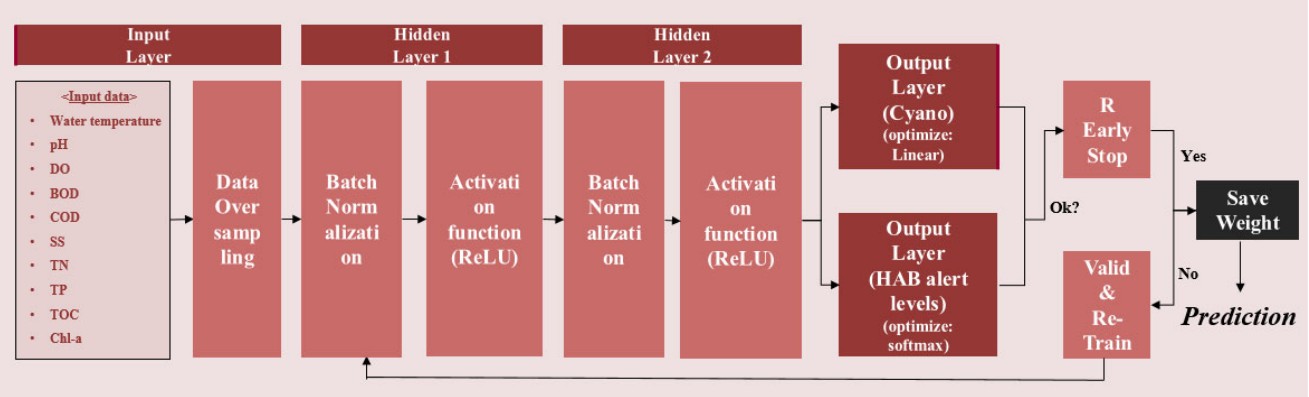

**Figure 4.** Deep learning model structure constructed in this study.

When SMOTE was added to the DNN algorithm, it showed satisfactory results in both training and validation (Table 3). Without the addition of SMOTE, the validation results for the HAB alert levels were very poor. It is considered that overfitting occurred due to a lack of training data for the HAB alert levels and the data biased towards level 0. It is believed that the validation performance improved as the training data increased and the data were more evenly distributed at each alert level by applying SMOTE to the DNN algorithm.

**Table 3.** Comparison of prediction performance.

| | | Cyanobacteria | | HAB Alert Levels | |
| --- | --- | --- | --- | --- | --- |
| | | DNN with SMOTE | DNN without SMOTE | DNN with SMOTE | DNN without SMOTE |
| Train | $R^2$ | 0.82 | 0.76 | 0.92 | 0.93 |
| | MAE | 0.5 | 0.7 | 0.1 | 0.1 |
| | RMSE | 0.7 | 0.8 | 0.1 | 0.1 |
| Validation | $R^2$ | 0.78 | 0.7 | 0.79 | 0.18 |
| | MAE | 0.6 | 0.7 | 0.1 | 0.2 |
| | RMSE | 0.7 | 0.9 | 0.2 | 0.3 |

### 3.4. Result of Cyanobacteria Prediction

HABs caused by the growth of cyanobacteria mainly occur in the summer season (between May and October) in Korea. Cyanobacteria are predicted during this period, and the results are used to manage HABs. Therefore, for the deep learning model to be utilized for managing HABs, it is crucial to accurately predict the number of cyanobacteria cells from May to October. The deep learning algorithm-based prediction model developed in this study demonstrates high prediction accuracy from May to October (Figure 5).

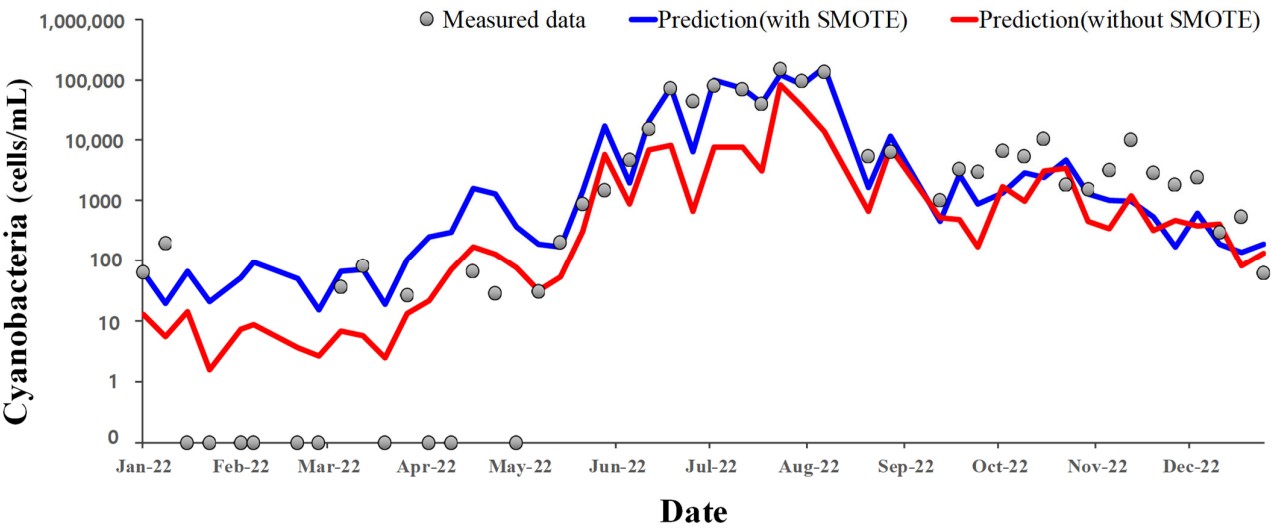

**Figure 5.** Result of the predicted number of cyanobacteria cells.

Although the predictive power may be somewhat lower during periods other than the summer season, the model has a high utility value as a HABs prediction model, as most cyanobacterial damage occurs during the summer season. In particular, the predictive accuracy of the model was very high for cyanobacteria cells above 10,000 cells/mL, which is the threshold for occurring cyanobacterial blooms. In light of these aspects, the model developed in this study is ideal for predicting and managing HABs.

### 3.5. Result of HAB Alert Levels Prediction

In the prediction results of HAB alert levels, even more, dramatic results were observed. When SMOTE was not applied to DNN, the prediction accuracy was high for levels 0 and 1 but relatively low for levels 2 and 3 (Table 4). It is speculated that this is due to the fact that the majority of the data are concentrated in levels 0 and 1. Therefore, the results indicate that there was insufficient training for levels 2 and 3 due to the relative scarcity of data for these levels. The poor prediction accuracy for levels 2 and 3 was improved by incorporating SMOTE within the DNN algorithm. Technologies such as deep learning models can produce optimal results when there are sufficient training data available. Therefore, when the amount of data for levels

2 and 3 was very limited, the predictive power for these levels was low. However, it was found that the predictive power for high-concentration cyanobacteria (levels 2 and 3) improved as the amount of data increased by applying SMOTE (Table 4). These results demonstrate that adding oversampling techniques such as SMOTE to deep learning algorithms for predicting imbalanced data such as HAB alert levels is a reasonable approach. The detailed prediction results are included in Table A1 of Appendix A.

**Table 4.** Accuracy of HAB alert levels prediction.

|  |  | Counts | | | Accuracy (%) | |
|---|---|---|---|---|---|---|
|  |  | Observation | Simulation without SMOTE | Simulation with SMOTE | Without SMOTE | With SMOTE |
| Level | 0 | 24 | 20 | 20 | 83.3 | 83.3 |
|  | 1 | 14 | 9 | 10 | 64.3 | 71.4 |
|  | 2 | 9 | 7 | 7 | 44.4 | 77.8 |
|  | 3 | 2 | 0 | 2 | 0.0 | 100.0 |

## 4. Discussion

In Korea, predicting the number of cyanobacteria cells has been used to respond to HABs. This is because the majority of HAB damage is caused by cyanobacteria. These predicted results are classified by the HAB alert system, and appropriate responses to the HABs are carried out according to the corresponding manuals for each level. This study proposed using a deep learning algorithm to predict both the HAB alert levels and the number of cyanobacteria cells in order to respond quickly to the occurrence of HABs. Various deep learning algorithms are being developed with recent technological advancements. Developing deep learning-based prediction technology is essential for responding to HABs. The predictive power of a deep learning model is significantly influenced by the number of input data. To accurately estimate HABs, securing as many high-quality data as possible is the most important challenge. The problem of imbalanced and insufficient data was addressed by combining the DNN prediction model with the SMOTE technique. It was observed that the predictive power of the insufficient data was significantly improved.

Despite the improved predictive power of the DNN prediction model combined with SMOTE, there are still issues that need to be addressed. Cyanobacteria data and water quality data that affect the growth and activity of cyanobacteria are measured on a weekly basis. However, the number of cyanobacteria cells varies greatly every day due to various conditions such as daily changes in weather, changes in water quality, and changes in discharge depending on the operation of hydraulic structures such as dams and weirs. Due to these characteristics, it is practically impossible to simply interpolate the data and create daily cyanobacteria data. If weekly data are used to input data, a deep learning model yields results only on a weekly basis. This is a significant limitation of data-based models such as deep learning models. Even with advanced deep learning algorithms such as long short term memory and transformer, it is impossible to solve this problem as long as there are limitations in input data. Even though the input data are obtained on a weekly basis, a physics-based model can be predicted on a daily basis due to its internal mechanism. Various conditions, such as weather, water quality, and discharge, that affect the growth and activity of cyanobacteria are incorporated into the internal mechanisms of physics-based models.

In this study, we proposed the following:

It is necessary to structurally combine the currently used physics-based and data-based models or to partially utilize the results of physics-based models to solve the limitations of data-based models, such as the lack of training data (Figure 6). Combining physics-based models with data-based models is a very effective approach because it maintains the accuracy of physical-based models while overcoming the limitations of data-based models.

This approach is used in various fields, such as AI, machine learning, and physics. One way to utilize some results from physics-based models is to modify or correct the output of the prediction model. However, when using these methods, the interaction between models is crucial. Physics-based models and data-based models cannot operate independently and must be interconnected. In addition, the inputs and outputs of the models must match, and the method of information transfer between the models must be clearly defined.

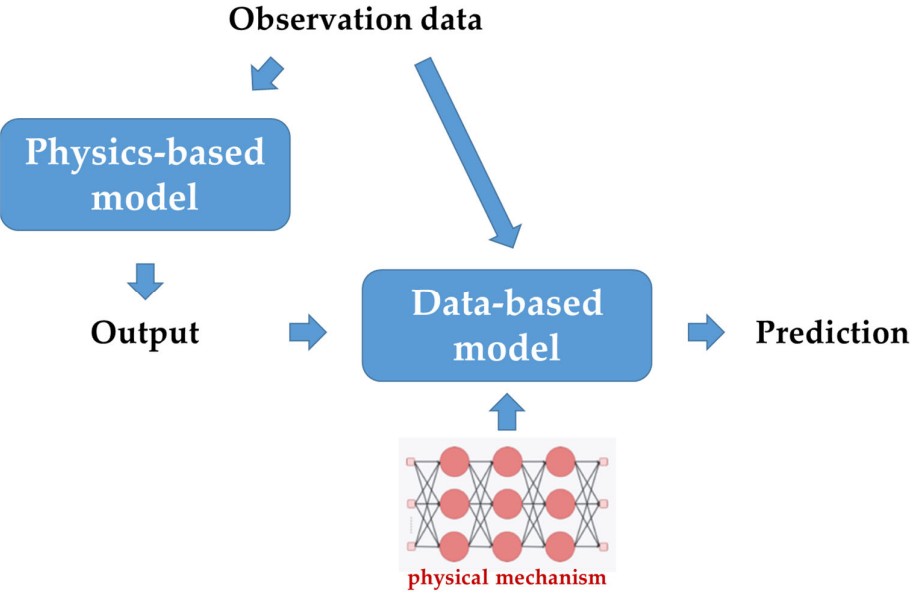

**Figure 6.** Example of an algorithm combining a physics-based model and a data-based model.

### 5. Conclusions

We improved the DNN algorithm to predict both the continuous data of the number of cyanobacteria and the categorical data of HAB alert levels simultaneously. The results of the improved DNN with SMOTE and the DNN without SMOTE prediction were compared and analyzed. The results of this study showed the applicability of the deep learning models using AI algorithms as a viable alternative to the traditional physics-based model for HAB prediction and management.

(1)  The DNN algorithm showed sufficient predictive power to estimate the number of cyanobacteria cells and HAB alert levels. It is essential to supplement the imbalanced data on cyanobacteria and HAB alert levels and secure the lacking data in order to ensure sufficient predictive power. The model combining the DNN algorithm with SMOTE showed better performance than the model that did not combine them.

(2)  High prediction accuracy was shown for the number of cyanobacteria cells during summer and for the serious alert level of HABs categorized as levels 2 and 3. In this aspect, it can be said that the developed HAB prediction deep learning model has sufficient applicability in responding to and managing HABs.

(3)  Algal blooms can be determined through algae concentrations and the HAB alert level. Therefore, these study-built deep learning models that can predict algae concentrations and HAB alert levels were used to prevent algal bloom-induced damage. To further advance this study, it is necessary to utilize the deep learning model developed in this study along with the existing physics-based model, EFDC-NIER, to respond to HABs.

**Author Contributions:** Conceptualization, J.K.; methodology, J.K.; software, J.K.; validation, J.K. and J.M.A.; formal analysis, H.K.; investigation, K.K.; resources, H.K. and K.K; data curation, J.K. and J.M.A.; writing—original draft preparation, J.K.; writing—review and editing, J.M.A.; visualization, J.K.; supervision, J.M.A.; project administration, J.K. and J.M.A.; funding acquisition, K.K. All authors have read and agreed to the published version of the manuscript.

**Funding:** This research was funded by the National Institute of Environmental Research (NIER), grant number NIER-2023-01-01-097.

**Data Availability Statement:** Not applicable.

**Acknowledgments:** This study was supported by a grant (NIER-2023-01-01-097) from the National Institute of Environmental Research (NIER), which is funded by the Ministry of Environment (MOE) of the Republic of Korea.

**Conflicts of Interest:** The authors declare no conflict of interest.

## Appendix A

Table A1 shows the predicted HAB alert levels, indicating the predicted probability for each level. It allows for comparison between the results with and without SMOTE applied.

**Table A1.** Result of the predicted HAB alert levels.

| Date | Observation | Predicted Probabilities for Each Level (%) | | | | | | | |
|---|---|---|---|---|---|---|---|---|---|
| | | Simulation without SMOTE | | | | Simulation with SMOTE | | | |
| | Level | 0 | 1 | 2 | 3 | 0 | 1 | 2 | |
| 3 January 2022 | 0 | 99.2 | 0.8 | 0.0 | 0.0 | 94.1 | 5.9 | 0.0 | 0.0 |
| 10 January 2022 | 0 | 99.9 | 0.1 | 0.0 | 0.0 | 89.5 | 10.5 | 0.0 | 0.0 |
| 17 January 2022 | 0 | 98.6 | 1.4 | 0.0 | 0.0 | 85.3 | 14.7 | 0.0 | 0.0 |
| 24 January 2022 | 0 | 100.0 | 0.0 | 0.0 | 0.0 | 99.9 | 0.1 | 0.0 | 0.0 |
| 3 February 2022 | 0 | 100.0 | 0.0 | 0.0 | 0.0 | 96.2 | 3.8 | 0.0 | 0.0 |
| 7 February 2022 | 0 | 99.8 | 0.2 | 0.0 | 0.0 | 96.5 | 3.5 | 0.0 | 0.0 |
| 21 February 2022 | 0 | 100.0 | 0.0 | 0.0 | 0.0 | 95.9 | 4.1 | 0.0 | 0.0 |
| 28 February 2022 | 0 | 100.0 | 0.0 | 0.0 | 0.0 | 98.9 | 1.1 | 0.0 | 0.0 |
| 7 March 2022 | 0 | 99.5 | 0.5 | 0.0 | 0.0 | 88.9 | 11.1 | 0.0 | 0.0 |
| 14 March 2022 | 0 | 100.0 | 0.0 | 0.0 | 0.0 | 99.7 | 0.3 | 0.0 | 0.0 |
| 21 March 2022 | 0 | 100.0 | 0.0 | 0.0 | 0.0 | 99.8 | 0.2 | 0.0 | 0.0 |
| 28 March 2022 | 0 | 99.9 | 0.0 | 0.1 | 0.0 | 96.6 | 3.4 | 0.0 | 0.0 |
| 4 April 2022 | 0 | 99.3 | 0.4 | 0.3 | 0.0 | 71.3 | 28.7 | 0.0 | 0.0 |
| 11 April 2022 | 0 | 99.9 | 0.1 | 0.0 | 0.0 | 92.9 | 7.1 | 0.0 | 0.0 |
| 18 April 2022 | 0 | 89.7 | 0.5 | 9.6 | 0.2 | 44.7 | 38.5 | 16.8 | 0.0 |
| 25 April 2022 | 0 | 96.5 | 3.5 | 0.0 | 0.0 | 95.0 | 4.9 | 0.1 | 0.0 |
| 2 May 2022 | 0 | 42.3 | 57.5 | 0.2 | 0.0 | 5.7 | 94.0 | 0.3 | 0.0 |
| 9 May 2022 | 0 | 98.6 | 1.4 | 0.0 | 0.0 | 95.3 | 4.0 | 0.7 | 0.0 |
| 16 May 2022 | 0 | 58.6 | 40.3 | 1.1 | 0.0 | 80.8 | 18.2 | 1.0 | 0.0 |
| 23 May 2022 | 0 | 75.2 | 18.6 | 5.4 | 0.8 | 74.8 | 21.7 | 3.5 | 0.0 |
| 30 May 2022 | 1 | 0.1 | 83.4 | 12.3 | 4.2 | 0.0 | 81.8 | 17.5 | 0.7 |
| 7 June 2022 | 1 | 1.0 | 96.7 | 1.8 | 0.5 | 12.2 | 56.6 | 31.2 | 0.0 |
| 13 June 2022 | 2 | 0.0 | 1.1 | 88.9 | 10.0 | 0.0 | 0.0 | 100.0 | 0.0 |
| 20 June 2022 | 2 | 0.0 | 0.1 | 90.2 | 9.7 | 0.0 | 0.0 | 100.0 | 0.0 |
| 27 June 2022 | 2 | 0.0 | 0.0 | 100.0 | 0.0 | 0.0 | 0.0 | 100.0 | 0.0 |
| 4 July 2022 | 2 | 0.0 | 0.3 | 99.2 | 0.5 | 0.0 | 0.0 | 100.0 | 0.0 |
| 13 July 2022 | 2 | 0.0 | 62.3 | 37.6 | 0.1 | 0.0 | 0.0 | 100.0 | 0.0 |
| 19 July 2022 | 2 | 0.0 | 87.2 | 7.6 | 5.2 | 0.0 | 0.0 | 100.0 | 0.0 |
| 25 July 2022 | 3 | 0.0 | 0.0 | 97.0 | 3.0 | 0.0 | 0.0 | 1.0 | 99.0 |
| 1 August 2022 | 2 | 0.0 | 72.0 | 28.0 | 0.0 | 0.0 | 11.6 | 42.6 | 45.8 |
| 8 August 2022 | 3 | 0.0 | 0.7 | 99.1 | 0.2 | 0.0 | 0.0 | 0.0 | 100.0 |

**Table A1.** *Cont.*

| Date | Observation | Predicted Probabilities for Each Level (%) | | | | | | |
|---|---|---|---|---|---|---|---|---|
| | | Simulation without SMOTE | | | | Simulation with SMOTE | | |
| | Level | 0 | 1 | 2 | 3 | 0 | 1 | 2 |
| 22 August 2022 | 1 | 7.0 | 93.0 | 0.1 | 0.0 | 0.6 | 99.4 | 0.0 | 0.0 |
| 29 August 2022 | 1 | 0.3 | 53.6 | 45.8 | 0.3 | 0.0 | 93.8 | 6.2 | 0.0 |
| 14 September 2022 | 0 | 38.3 | 61.7 | 0.0 | 0.0 | 9.5 | 90.5 | 0.0 | 0.0 |
| 20 September 2022 | 1 | 38.8 | 61.2 | 0.0 | 0.0 | 11.7 | 88.3 | 0.0 | 0.0 |
| 26 September 2022 | 1 | 96.0 | 3.9 | 0.1 | 0.0 | 17.5 | 40.1 | 42.4 | 0.0 |
| 4 October 2022 | 1 | 0.3 | 97.6 | 1.3 | 0.8 | 0.2 | 94.1 | 5.7 | 0.0 |
| 11 October 2022 | 1 | 75.8 | 17.1 | 0.7 | 6.4 | 0.3 | 0.5 | 99.0 | 0.2 |
| 17 October 2022 | 2 | 20.2 | 79.4 | 0.2 | 0.2 | 12.9 | 86.7 | 0.1 | 0.3 |
| 24 October 2022 | 1 | 28.1 | 8.5 | 61.8 | 1.6 | 3.0 | 3.2 | 93.7 | 0.1 |
| 31 October 2022 | 1 | 65.8 | 34.2 | 0.0 | 0.0 | 43.9 | 52.5 | 3.6 | 0.0 |
| 7 November 2022 | 1 | 52.5 | 47.5 | 0.0 | 0.0 | 50.2 | 49.8 | 0.0 | 0.0 |
| 14 November 2022 | 2 | 15.4 | 84.6 | 0.0 | 0.0 | 0.1 | 34.6 | 65.3 | 0.0 |
| 21 November 2022 | 1 | 35.6 | 64.4 | 0.0 | 0.0 | 48.8 | 51.2 | 0.0 | 0.0 |
| 28 November 2022 | 1 | 41.0 | 59.0 | 0.0 | 0.0 | 43.8 | 56.2 | 0.0 | 0.0 |
| 5 December 2022 | 1 | 29.0 | 71.0 | 0.0 | 0.0 | 41.6 | 58.4 | 0.0 | 0.0 |
| 12 December 2022 | 0 | 32.0 | 68.0 | 0.0 | 0.0 | 26.5 | 73.5 | 0.0 | 0.0 |
| 19 December 2022 | 0 | 36.5 | 63.5 | 0.0 | 0.0 | 36.9 | 63.1 | 0.0 | 0.0 |
| 26 December 2022 | 0 | 57.7 | 42.3 | 0.0 | 0.0 | 69.5 | 30.5 | 0.0 | 0.0 |

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
