# Peer review of "Research on the Development and Application of a Deep Learning Model for Effective Management and Response to Harmful Algal Blooms"

_water, doi:10.3390/w15122293_

Round 1

Reviewer 1 Report

The manuscript “Research on the Development and Application of Deep Learning Model for Effective Management and Response to Harmful Algal Blooms” presents an interesting approach on algal biomass. Following points may further improve it;

1.     Include quantitative outcome of the study in the abstract.

2.     Include one or two more keywords to improve visibility of the study.

3.     The first section may be linked to global causes like SDGs/net zero targets. You may refer the latest study – “A novel optimization approach for biohydrogen production using algal biomass”.

4.     Avoid too much lumping of references like [2-6].

5.     Why DNN was chosen.  You may refer for this - Weather Impact on Solar Farm Performance: A Comparative Analysis of Machine Learning Techniques.

6.     How was uncertainty quantified in this study?

7.     Section 2.4 is not cited.

8.     In Table 2, check subscripts in R2.

9.     Figure 6 is not clear.

10.  Ensure all the abbreviations are explained.

11.  Kindly check the typos in the paper.

12.  The study may be improved with latest and relevant studies like 10.3390/w13182457 and 10.1021/acs.energyfuels.2c01006.

It is a well written and compact paper. It may be considered after major revision.    

Kindly check for typos and subscripts.  

Author Response

Thank yor for your review. Please see the attachment.

Reviewer 2 Report

The work is a valuable contribution to the development of an early warning system for the occurrence of harmful algal blooms (HABs). An early warning system for HABs serves as a food safety intervention tool to identify potential risks required to be managed. In this study DNN algorithm was improved to predict both the continuous data of the number of cyanobacteria and the categorical data of HAB alert levels simultaneously. The results of the improved DNN with SMOTE and the DNN without SMOTE prediction were compared  with each other and with the measured number of cells and analysed.

Author Response

Thank you for your review. We are continuously advancing our research

Reviewer 3 Report

Line 13: syfstem → system

Abstract: No information given about the model accuracy in terms of statistical indices such as R2, RMSE, and so on.

Lines 25-26: Do you mean the “The Four Major Rivers Restoration Project of South Korea”? Can you please add more information? I’m not from Korea. But, I guess this project caused frequent HAB events in the major Korean Rivers.

Lines 29-30: I think extreme rainfalls also play a key role in HABs in Korean rivers since summer is the wet season in Korea. If so, sediment resuspension due to changes in hydrodynamic conditions in rivers can re-suspend the bed sediments that are home for large nutrient loads in the rivers (see, “Uncertainty quantification of granular computing-neural network model for prediction of pollutant longitudinal dispersion coefficient in aquatic streams” for justification of your statement).

Lines 45-67: Although the authors acknowledge the wide application of AI-based models in water resources, they are failed to properly support this section with appropriate references. I suggest the authors to, at least, add some AI-based models reported for water quality prediction in riverine systems such as “Estimation of the dispersion coefficient in natural rivers using a granular computing model”.

Introduction: Eutrophication is a main concern in surface water systems, even the world’s largest inland lakes (see, “Caspian Sea is eutrophying: The alarming message of satellite data). I would like some statements that highlight the impact of eutrophication in different water bodies around the world from rivers to lakes, reservoirs, ponds, and wetlands.

Line 103: biochemical oxygen demand → the 5-day biochemical oxygen demand

Section 2.2: Did you use the measured data in a sampling point? Or more sampling points?

Section 2.2: Please add a tables, showing the main statistical indices of the data used for training and validation of the DNN model.

Lines 127-130: Why not TN? TN may contribute to eutrophication even more than DO.

Figure 3 is not informative. Please remove or elaborate it with your input data.

All equations 1 to 5 are relevant. Please remove them and add some references instead of (e.g., “Artificial Neural Networks and Regression Analysis for Predicting Faulting in Jointed Concrete Pavements Considering Base Condition”).

Line 172: Why not between -1 and 1?

Section 3.2: This section has poorly supported by appropriate references.

Good luck!

Reviewer 4 Report

Dear authors,

you have written a nicely manuscript that was easy to read.

HABs can be a risk for humans (e.g. shellfish, fish and drinking water), and thus the topic is a relevant one.

But I have four major concerns that should be addressed and in addition three minor comments:

Major concerns:

1) In the introduction authors mention that "AI-driven deep learning models....and demonstrate excellent prediction performance in diverse fields such as climate,.....and water quality".

To be objective, I think authors should also mention that there have been several examples were AI driven (deep learning) models turned out to be drastic and have had bad prediction performnces and even bad consequences for humans.

2) Table 3 contains a mistake, i.e. the level 2 accuracy Without SMOTE is given at 44.4%, this is incorrect. It has to be 77.8%. As a result, there is only a clear difference between Without and With SMOTE at level 3, but not at levels 0, 1 and 2. So the current conclusion is also incorrect, i.e. that With SMOTE is better in the whole range.

3) When I compare the "simple" data given in Figure 2, where Water temperature is shown to correlate nicely with the number of Cyanobacteria, to the "complex" AI-based generated model predicted number of Cyanobacteria and Measured number of Cyanobacteria in Figure 5, I dare to conclude that a simple prediction by only measuring the watertemperature is just as good. This means there is no need for an AI tool.

4) Based on comment 3, would an AI-based deep learning tool to predict the watertemperature a week ahead and thus predict the number of Cyanobacteria not be a far better model to protect consumers of drinking water?

Minor comment:

5) In the Materials and Methods part, authors mention that data from 2012 to 2021 were used for training and testing, and the data from 2022 were used to evaluate the prediction accuracy of the constructed model. Did authors think about testing the influence of using different/others years for training and testing and selection another year(s) to evaluate the precidiction accuracy?

6)   Table 2 is unclear. In the text authors mention: "...it showed satisfactoty results in both training and validation (Table 2)." But table 2 says Train en Test. Moreover, the numbers, 1.0 means 100% and e.g. 0.82 means 82% or is it an R value that is given?

 7) Referencing should be improved, e.g. only 17 references mentioned, some already of previous studies by the authors. Include references for studies were AI did not work (see major comment 1). Also include references proving that the number of cyanobacteria correlate strong with the presence of toxins, as otherwise a model for predicting numbr of cyanobacteria is useless to protect consumers.

English is fine

Round 2

Reviewer 1 Report

The authors have done well in revising the manuscript. It may be accepted. 

Reviewer 3 Report

Thank you for addressing my comments. My suggestion is acceptance. Congratulations.